# Association between African Dust Transport and Acute Exacerbations of COPD in Miami

**DOI:** 10.3390/jcm9082496

**Published:** 2020-08-03

**Authors:** Miguel Pardinas Gutierrez, Paquita Zuidema, Mehdi Mirsaeidi, Michael Campos, Naresh Kumar

**Affiliations:** 1Pulmonary and Critical Care, Jackson Memorial Hospital, Miami, FL 33136, USA; mpardinasg@gmail.com (M.P.G.); msm249@miami.edu (M.M.); 2Rosenstiel School of Marine and Atmospheric Science, University of Miami, Miami, FL 33149, USA; pzuidema@miami.edu; 3Division of Pulmonary, Allergy, Critical Care and Sleep Medicine, University of Miami School of Medicine, Miami, FL and Miami Veterans Affairs Medical Center, Miami, FL 33136, USA; 4Environmental Health Division, Department of Public Health Sciences, University of Miami School of Medicine, Miami, FL 33136, USA

**Keywords:** Saharan dust outbreak, COPD, climate change, particulate matter, extreme weather

## Abstract

Background: Air pollution is increasingly recognized as a risk factor for acute exacerbation of chronic obstructive pulmonary disease (COPD). Changing climate and weather patterns can modify the levels and types of air pollutants. For example, dust outbreaks increase particulate air pollution. Objective: This paper examines the effect of Saharan dust storms on the concentration of coarse particulate matter in Miami, and its association with the risk of acute exacerbation of COPD (AECOPD). Methods: In this prospective cohort study, 296 COPD patients (with 313 events) were followed between 2013 and 2016. We used Light Detection and Ranging (LIDAR) and satellite-based Aerosol Optical Depth (AOD) to identify dust events and quantify particulate matter (PM) exposure, respectively. Exacerbation events were modeled with respect to location- and time-lagged dust and PM exposures, using multivariate logistic regressions. Measurements and main results: Dust duration and intensity increased yearly during the study period. During dust events, AOD increased by 51% and particulate matter ≤2.5 µm in aerodynamic diameter (PM_2.5_) increased by 25%. Adjusting for confounders, ambient temperature and local PM_2.5_ exposure, one-day lagged dust exposure was associated with 4.9 times higher odds of two or more (2+ hereto after) AECOPD events (odds ratio = 4.9; 95% CI = 1.8–13.4; *p* < 0.001). Ambient temperature exposure also showed a significant association with 2+ and 3+ AECOPD events. The risk of AECOPD lasted up to 15 days after dust exposure, declining from 10× higher on day 0 to 20% higher on day 15. Conclusions: Saharan dust outbreaks observed in Miami elevate the concentration of PM and increase the risk of AECOPD in COPD patients with recurring exacerbations.

## 1. Introduction

Acute exacerbations of chronic obstructive pulmonary disease (AECOPD) are events that significantly alter the course of the disease by accelerating lung function decline, negatively impacting quality of life and increasing mortality [1,2,3]. It is unclear why AECOPD are particularly frequent events for some subjects and not for others. Although genetic risk factors are being actively sought [4], it is imperative to not only focus on genetic and epigenetic factors, but also on the environmental factors that influence them.

Epidemiological studies suggest an increased risk of exacerbations during high air pollution episodes [5,6,7,8], accounting for up to 9% of hospital admissions for AECOPD [9]. Among air pollutants, inhalation of particulate matter (PM) ≤2.5 µm and ≤10 µm in aerodynamic diameter (PM_2.5_ and PM_10_, respectively) is linked to altered immunity, oxidative stress, as well as airway and systemic inflammation [10,11]. In urban areas, the main sources for these pollutants are anthropogenic, including the combustion of fossil fuel and biomass burning [12]. However, the mechanical breakdown of Earth′s crust, as well as the regional and global transport of desert dust, contributes to PM exposure as well. These dust events are becoming more frequent in some regions with intensifying global warming and desertification [13]. African dust is of particular interest not only because it accounts for half of the total dust emission [14,15], but also because transported dust has been frequently observed across multiple continents, including Europe [16] and the Americas [17]. The dust event is characterized by a high concentration of coarse particles (PM_2.5–10_) that travel far distances, typically crossing the Atlantic in approximately 10 days [15], and they may circumnavigate the globe within 13 days at higher latitudes on occasion [18]. Air sampling, conducted in the Caribbean and South Florida since the 1970s, has clearly documented elevated levels of PM during dust events [19,20,21], with most of the transported dust particles being ≥2.5 µm and ≤10 µm in aerodynamic diameter (PM_10–2.5_), and the remaining fraction of particles are ≤2.5 µm [21]. During dust events, levels of PM_10_ exceed the WHO threshold of good air quality (50 µg/m^3^/24 h) in the Caribbean and South Florida [22].

Although there is overwhelming epidemiological and toxicological evidence for the adverse health effects of PM_2.5_ and PM_10_ on pulmonary and cardiovascular diseases [23,24], including AECOPD [8], only a few studies provide insight into the adverse health effects of transported dust in the US. Most studies document the effects of dust events that occur in Asia [25,26,27], but none have evaluated the impact of African dust transported to America on AECOPD. Leveraging a well-characterized COPD cohort, air pollution data and dust data, this paper examines the risk of AECOPD with respect to time-lagged exposure to transported dust, adjusting for exposure to local air pollution and confounders including ambient temperature and smoking status.

## 2. Materials and Methods

### 2.1. Study Site

This study included patients from the Miami catchment area, defined as an area located within the geographic range of 81° W to 80° W and 25° N to 27.15° N, because 91% of the COPD patients enrolled in the study lived within the catchment area.

### 2.2. Study Design and Subjects

We used a prospective clinical cohort design to recruit patients. COPD patients were enrolled during their pulmonary clinic visits to the Miami Veterans Affairs (VA) Medical Center between 2013 and 2016. COPD was defined as the presence of airflow obstruction in spirometry (FEV_1_/FVC < 0.7 (or lower limit of the normal) after administrating a short acting bronchodilator). COPD patients without any episode of exacerbation during the clinic visit were offered an opportunity to participate in the study. They were followed every 6 months for their regular COPD care. During their clinic visit, their medical records were reviewed for any episodes of emergency room visit, hospitalization and outpatient visits to identify all AECOPD episodes. AECOPD events were defined by an increase in respiratory symptoms that required administration of systemic steroids, antibiotics, or both. All events were defined as cases and subjects who did not experience any AECOPD events during the study period as controls. For each AECOPD event, date, smoking status and date of symptom onset were recorded after the chart review. This study was approved by the Miami VA IRB (IRBNet ID 1161589-9).

### 2.3. Environmental Data

PM_2.5_ exposure assessment: PM_2.5_ data were not available at/around the place of residence for any of the subjects. We used a hybrid approach to compute location- and time-lagged PM_2.5_ exposure. This approach used ground monitored PM_2.5_ data from Environmental Protection Agency (EPA) sites within Florida, satellite data from NASA and meteorological data from the National Climatic Data Center (NCDC). Hourly ambient PM_2.5_ data were acquired from the EPA for all sites in Florida within 84° W to 79° W and 24.5° N to 29.2° N from 2012 to 2016 [28]. This extent was included for two different reasons. First, the transported Saharan dust impacts most parts of Florida. Second, there are only two sites in Miami-Dade County where hourly PM_2.5_ is monitored. Data from multiple sites were needed to develop an empirical relationship between satellite-based Aerosol Optical Depth (AOD) and PM_2.5_ monitored on the ground, in order to extrapolate location and time–space estimate of PM_2.5_ for the entire study area.

MODerate Resolution Imaging Spectroradiometer (MODIS) data, onboard NASA′s Terra and Aqua satellites, were used to compute AOD, which was needed in order to determine daily PM_2.5_ exposure at the place of residence of each subject. A total of 8.9 million AOD values were computed at 3 km spatial resolution from 2012 to 2016 [29,30]. Global surface hourly meteorological data, such as temperature, humidity, wind velocity and atmospheric sea level pressure, were acquired from the NCDC. An empirical relationship between in situ monitored PM_2.5_ and AOD was developed, adjusting for meteorological conditions. A unit of AOD corresponded with ~21.4 µg/m^3^. Extrapolating this relationship, PM_2.5_ was estimated for each day and location wherever AOD was computed using satellites (see Sinha and Kumar [31] for details), and results of the empirical model are shown in Table 1 and Table 2 and Figure 1 and Figure 2. Using the local time–space kriging (LTSK) [32], daily PM_2.5_ exposure was computed for 15 days before each AECOPD event, and daily exposure for up to 15 days before a clinic visit for controls.

Dust exposure assessment: Days of Saharan dust event and dust intensity were estimated using a Light Detection and Ranging (LIDAR) instrument [34,35], which allowed us to develop a vertical profile of aerosols and transported dust (Figure 2). In situ PM_10_ data monitored by gravimetric and photometric methods were used to identify dust days, and the LIDAR volume depolarization ratio was used as a proxy of dust intensity (Figure 2). Since most parts of South Florida are affected by transported dust similarly on a given day, all events and controls were considered as exposed during those days. Daily dust exposure was computed for 15 days before the exacerbation date for AECOPD events and clinic exam date for controls who did not have any AECOPD events during the study period. Distributive inverse time weighted lagged dust exposure (*X_it_*_|*l*|_) for *ith* subject on *tth* day, and *lth* lag day, before the exacerbation was computed as
*X_it|l|_* ~ Σ*_lϵL_*(*D_|t−l|_* . (1/*l*))/Σ*_lϵL_* (1/*l*)(1)
where *D*_|*t−l*|_ is dust intensity on the *lth* day before the exacerbation event for the cases, and before the clinical exam day for controls; *l* is time 1,2,3,…,L, and L = 15 days.

Analytical Method. The analytical strategy builds on the following assumptions: (i) ambient exposure to Sub-Saharan dust (*X_it_*) is likely to be same for all subjects on a given day because transported dust is part of the large air masses; (ii) ambient daily PM_2.5_ exposure is likely to be unique for each subject at his/her place of residence due to spatiotemporal heterogeneity in PM_2.5_ sources; (iii) smoking, age, severity of lung function and ambient temperature can confound the risk of AECOPD; and (iv) severity and patterns of AECOPD are likely to be unique for each patient. Risk of AECOPD was modelled using a multivariate logit model with standard error adjusting for intragroup correlation for multiple events for some patients (i.e., logit function with vce(cluster) option) in STATA version 14.2 [36].

## 3. Results

### 3.1. Subjects and AECOPD Prevalence

A total of 296 unique COPD patients were recruited. Between 2013 and 2016, 34% (or 101) had one or more episodes of AECOPD. These subjects were of comparable age, but had lower lung function, as detailed in Table 3. Only six of them were women (and four of them did not have an exacerbation). All had a smoking history (43.2% active smokers at study entry). For the analysis, we excluded subjects who did not reside in the Miami catchment area (28 controls and 16 AECOPD events), leaving a total of 297 AECOPD episodes and 166 controls (Table 4).

### 3.2. PM_2.5_ Exposure

The annual average concentrations of PM_2.5_, AOD and meteorological conditions in the study area are presented in Table 3. The daily average PM_2.5_ concentration was 10.5 µg/m^3^ between 2013 and 2016, below the EPA threshold (12 µg/m^3^) [37]. There was a strong seasonal trend in PM_2.5_ and AOD distribution, with both peaking during the summer and fall seasons, which coincided with the dust outbreak season (Figure 3).

The concentration of PM_2.5_ did not vary significantly between the Miami catchment (10.5 µg/m^3^) and the rest of Florida (9.5 µg/m^3^), excluding dust days, but the AOD concentration in the Miami catchment was significantly higher than that outside the catchment area (0.168 in Miami catchment versus 0.13 outside the catchment; difference ~0.038; *p* < 0.0001) (Table 5). The differences in the meteorological conditions inside and outside the Miami catchment area were not statistically significant, but a deeper, more humid atmosphere above Miami could generate aerosol swelling that may explain a higher AOD above Miami, as evident from the slightly higher dew points (Table 5). The ambient surface temperature was also slightly higher during events (22.14 °C during dust events versus 20.96 °C during non-dust events; difference ~1.17 °C; *p* < 0.001).

### 3.3. PM_2.5_ and AOD during the Dust Outbreak 

Dust events were grouped and their intensity graded based on the depolarization ratio from LIDAR. Both PM_2.5_ and AOD were significantly higher during dust events as compared to non-dust event days (Table 5 and Figure 4). For example, PM_2.5_ concentration was 20% higher during the dust events, and AOD increase was 37.7% during the low dust events (i.e., when the depolarization ratio was 0.1 or less) in the entire study area, but it increased to 60.1% during medium and high dust events in the entire region. The average AOD during medium and high dust events in the Miami catchment was >0.24 as compared to 0.162 during no-dust events (mean difference ~0.081; *p* < 0.001). A similar pattern was observed in areas outside the catchment, suggesting most parts of Florida experience a similar exposure to sub-Saharan dust (Table 6).

### 3.4. Dust Outbreak Duration and Intensity

Most dust outbreaks occur during the summer and fall months (June to September), and the LIDAR data suggest an increase in dust duration as well as dust intensity from 2013 to 2016 (Table 6 and Figure 2). For example, the average number of days with significant dust increased from 29 in 2012 to 68 in 2016, and the average depolarization ratio increased from 0.13 in 2013 to 0.16 (difference ~0.03; *p* < 0.001). With the exception of 2013, the number of dust event days has increased from June to September between 2014 and 2016, suggesting expanding dust events over the fall seasons (Figure 5).

### 3.5. AECOPD by PM_2.5_ and Dust Exposure

Ambient one-day lag PM_2.5_ exposure (at the place of residence) was ~15 μg/m^3^ for cases (i.e., 1 or more AECOPD event) as compared to ~14 μg/m^3^ for controls. Among subjects who experienced 2+ AECOPD during the study period, the one-day lag PM_2.5_ exposure was 18 ug/m^3^. There were no controls on the days of first AECOPD during the dust event days. Thus, it was not feasible to compute odds ratios for first AECOPD events. Thus, analyses were restricted to 2+ and 3+ AECOPD events. The results of the logistic regression, adjusted for age, smoking, asthma status, lung function and ambient temperature, are shown in Table 7a. Among all confounders, age and smoking showed significant associations with AECOPD in some of the models, but not all. The risk of 2+ AECOPD increases by 14% with a unit increase in one-day lag PM_2.5_ exposure (odds ratio ~1.14; 95% confidence interval CI = 1.08–1.19; *p* < 0.001; Model 1). The one-day lag ambient temperature also showed a positive association with 2+ AECOPD (odds ratio ~1.12; 95% CI = 1.033–1.217; *p* < 0.01 (Model 2)). The odds of 2+ AECOPD was 5.7 times higher for one-day lag dust exposure without adjusting for ambient temperature or local PM_2.5_ exposure (odds ratio ~4.1; 95% CI = 2.08–16.01; *p* < 0.001 (Model 3)). When adjusted for PM_2.5_ exposure and ambient temperature, the odds of 2+ AECOPD was 4.9 times higher as compared to non-dust event days (odds ratio ~4.95; 95% CI = 1.82–13.45; *p* < 0.001; (Model 6)). The trend for subjects who experienced 3+ AECOPD during the study period remained the same, but the odds ratio was lower (Table 7b). However, ambient temperate, independently and along with PM_2.5_ and dust exposure, showed a strong significant association with 3+ AECOPD (Table 7b). For example, odds of 3+ AECOPD was 21% higher with a unit increase in ambient temperature when the effects of PM_2.5_ and dust exposure are accounted for (odds ratio ~1.21; 95% CI = 1.096–1.345; *p* < 0.01 (Model 6)). Among all the confounders included in the analysis, only age showed a marginally significant association with the AECOPD.

### 3.6. AECOPD and 15-Day Time-Lagged Dust Exposure

Daily time-lagged dust exposure was computed for 15 days before all AECOPD events, and 15 days before clinic visits for controls. Dust exposure was conceptualized in three different ways, as follows: (a) binary exposure—exposed if the dust event was observed on the lagged day, unexposed otherwise; (b) dust intensity measured by LIDAR for each lagged day; and (**c**) distributive cumulative dust intensity, which is an inversely time weighted average of dust intensity for a given time lag. The odds of 2+ AECOPD and 3+ AECOPD were modeled separately with respect to all three measures of daily lagged dust exposure (Table 8).

All three measures of daily lagged dust exposure were associated with both 2+ AECOPD and 3+ AECOPD for up to 15 days (Figure 6). For example, odds of 2+ AECOPD and 3+ AECOPD were 28% and 20% higher (even for the 15th day distributed lagged dust exposure), respectively (for 2+ AECOPD: odds ratio = 1.28; 95% CI = 1.09–1.51; *p* < 0.001; for 3+ AECOPD: odds ratio = 1.21; 95% CI = 1.07–1.36; *p* < 0.001; Table 8). For two other dust exposures, the odds of AECOPD varied with the increase in time lag. For example, 2+ AECOPD were 10× more likely to occur with a unit increase in dust intensity on the same day of the event, which gradually declined up to five days, and became 11× by six-day lag (Figure 6). A similar pattern was observed for 3+ AECOPD. A dust exposure that occurred 15 days before the AECOPD event was associated with 5.5× higher odds of 3+ AECOPD (odds ratio ~5.5; 95% CI = 1.57–19.3; *p* < 0.001).

## 4. Discussion

This study shows a significant increase in AOD as well as PM_2.5_ during the Saharan dust events in Florida. The time-lagged dust exposure was associated with an increase in the risk of AECOPD among COPD patients, and this risk lasted for weeks after the exposure. PM_2.5_ exposure and ambient temperature also showed independent associations with the risk of COPD exacerbation (for both 2+ and 3+ AECOPD) events. These findings are consistent with the emerging body of literature that shows a direct link between dust exposure and increase in the risk of different diseases, including pneumoconiosis and granulomatous diseases [14] and asthma exacerbations [38,39]. A study conducted in Taiwan showed a 20% increase in COPD-related hospital admissions in response to dust exposure [40]. A recent study also documents a 3.6-fold increase in COPD exacerbations with the increase in PM_10_ during four dust storms [41].

Dust events are likely to become more frequent and intense with changing climate and weather patterns [14]. Dust storms originate from drylands, which constitute approximately 40% of the world’s surface and account for 30% of the world population [14,42]. Dust impacts are not restricted to local areas, because airborne dust can be transported thousands of miles away from its origin source. The largest source of atmospheric dust is the Sahara-Sahel region in North Africa. The dust originating from this region reaches different parts of the world, including the Americas across the Atlantic Ocean [43,44]. In the US, the greatest number of dust storms occur in western states (Arizona, California, Washington and Nevada), particularly during the early afternoon hours of summer months [45]. This study provides epidemiological evidence of the association between AECOPD and exposure to the transported dust originating on another continent.

In this study we observed that an increase in ambient temperature was associated with an increased risk of AECOPD, which is inconsistent with the literature. Generally, a low temperature (especially very cold) that constricts airways is shown to increase the risks of AECOPD [46,47]. However, temperatures in Miami remain high year-round. Thus, the effects of high temperature on AECOPD are likely to be indirect, because elevated temperatures keep airborne particles afloat for longer, which can result in elevated PM_2.5_ exposure. The current study suggests that both PM_2.5_ and ambient temperature were associated with an increased risk of AECOPD.

Urban residents are chronically exposed to PM_2.5_ and PM_10_ from local sources, but their PM exposure further increases during dust events [27]. PM_10_ concentrations can exceed 1000 μm/m^3^ during dust storms, and may surpass 15,000 μm/m^3^ during severe events [48,49]. These high levels of PM_10_ during dust storms are associated with respiratory diseases [50]. Moreover, dust is also associated with the risk of other diseases, including pulmonary infectious [51,52] and cardiovascular diseases [53]. PM has been shown to directly contribute to all-cause and cause-specific mortality as well [54]. For example, a study in Barcelona showed that a daily increase of 10 μg/m^3^ in PM_10–2.5_ from Saharan dust was associated with an 8.4% increase in mortality [55].

The findings of this research have implications for the management of dust exposure and its associated health risks. First, there is overwhelming evidence of the intensifying of dust outbreaks worldwide, including in Asia, Africa and America, over the past 30 years. The highest dust intensity in Miami was observed in 1983 after the intense El Niño events. Thus, a changing climate, that affects the phasing of the North Atlantic Oscillation, could intensify dust storms in the future [38] and elevate exposure to coarse particles (PM_10–2.5_). Second, dust exposure synergistically magnifies the effects of local pollutants, due to its enhanced toxicity [26,56]. Transported dust is rich in minerals, such as iron, which can enhance the growth and viability of microbial communities, and dust particles also serve as a transport agent for microbial agents [57,58]. Third, dust storms impact large areas and have a widespread effect. For example, Saharan dust affects most parts of southeastern US, including Florida and Texas [59]. Fourth, transported dust may not only impact respiratory diseases, but other diseases as well, including allergies and immunological disorders. Therefore, intensifying dust exposure is likely to have a heavier effect on human health, which warrants proactive dust exposure management strategies to mitigate these adverse health effects.

Multiple stakeholders need to be engaged in managing the health risks of dust exposure. First, real-time surveillance of dust transport, and its associated region-, population- and disease-specific and time-lagged health risks, is needed. As demonstrated in this paper, a hybrid approach that integrates satellite-based AOD, in situ monitoring of the vertical profile of dust and particulate air pollution data can be used to develop location- and time-specific estimates of transported dust. Second, age-, gender- and disease-specific risk assessments are warranted as there is limited literature on the biological impact of the imported dust. Third, healthcare professionals need to be trained concerning the health risks of dust exposure and its persistence for days and weeks. They need to interrogate their patients regarding their recent dust exposure, assess its potential health risks and provide them with information on dust exposure avoidance. Finally, it is important to provide the general public with information regarding the real-time health risks of dust exposure through multi-media platforms, such as weather news and cellphone application(s) that provide information on the transported dust concentration and its associated health risk.

Assessing the health risk of short-term pollution events merits researchers′ attention. However, it has potential methodological challenges. Therefore, the findings of this research must be interpreted considering the following weaknesses. First, the focus of this study on veterans with COPD constraints the scope of its generalizability. Second, the data may be subject to recall biases, given that patients were seen every 6 months. However, performing the study in our cohort minimized this risk, as it is considered a “captive” population with most subjects receiving their healthcare at one place, and they visit VA, which has systematic records of each clinical visit which we used for verification. Third, the study controlled for a limited number of confounders due to limited data on comorbidities and other risk behaviors. If we consider the dust event as a natural experiment, then the patterns of comorbidities and other risk factors are unlikely to change before, during and after the dust events. Fourth, there is the plausibility of exposure misclassification for some, because dust exposure was considered as being the same for all subjects on a given day. There can be individual differences in the time participants spend indoors and outdoors. However, their time-activity pattern is unlikely to change dramatically before, during and after the dust events, unless they are aware of “outdoor dust exposure” and take precautions to reduce their exposure. Fourth, the quantification of the LIDAR data can be subject to bias. Nonetheless, the dust intensity readings obtained from LIDAR showed a positive association with satellite-based assessments of AOD in Miami (>0.2 during the dust event).

## 5. Conclusions

The literature on the health effects of dust exposure is scant, and further research is warranted to build up the empirical evidence of the disease-specific effects of dust exposure and its persistence over time. In particular, understanding the environmental risks of AECOPD exacerbation is important, given its potential for improving clinical recommendations regarding individual behavior modification during discrete dust and/or pollution events. There is a need to develop strategies to engage multiple stakeholders, in order to manage the adverse health effects of dust exposure, ranging from developing real-time disease-specific health risks to training healthcare professionals in assessing the health risk of time-lagged dust exposure among their patients, so as to engage them in dust exposure avoidance.

## Figures and Tables

**Figure 1 jcm-09-02496-f001:**
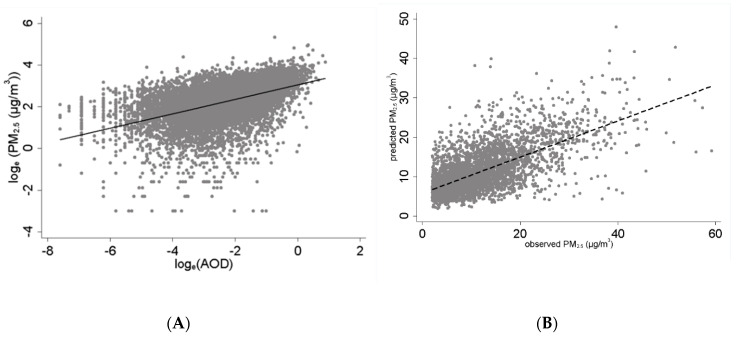
Association between measured PM_2.5_ with AOD and predicted PM_2.5_ values during the study period. (**A**) Association between airborne PM_2.5_ and AOD in South Florida, 2012–2016. (**B**) Predicted versus observed PM_2.5_ in South Florida, 2012–2016. AOD and PM were collocated if their time stamps were within ±0.75 h and located within 0.025° (or ~2.4 km) geographic distance.

**Figure 2 jcm-09-02496-f002:**
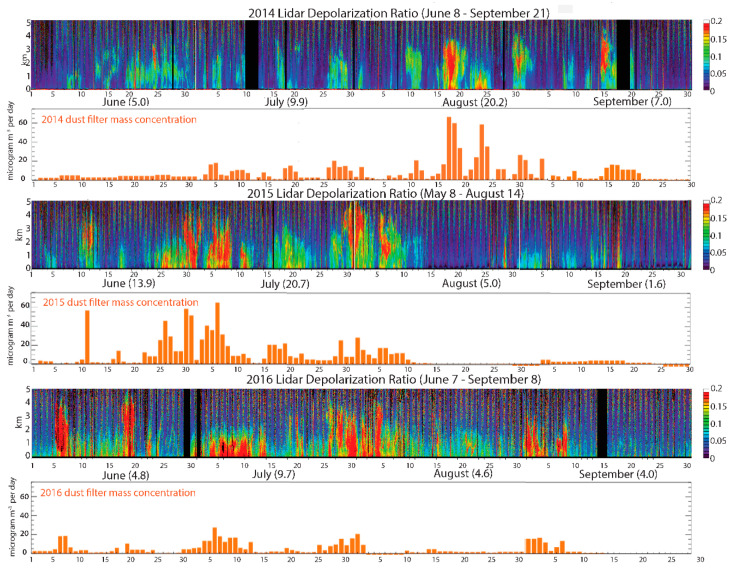
Dust events recorded between 2014 and 2016. LIDAR data coupled with in situ monitored PM_10_ using both gravimetric and photometric methods were used to identify dust days, and depolarized ratio was used as a proxy of dust intensity. The colored graphs show the LIDAR readings, with higher depolarization ratios are shown in red. The bar graphs show the corresponding increases in dust mass recorded by weighing air filters [33].

**Figure 3 jcm-09-02496-f003:**
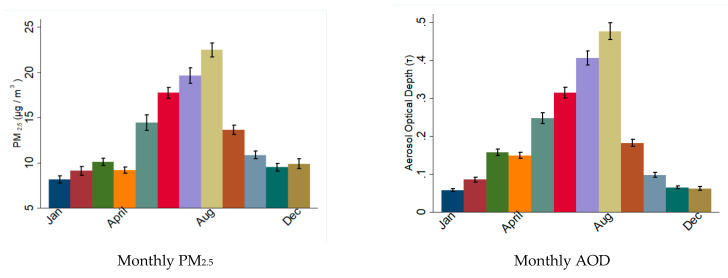
Monthly distribution of particulate material during the study period in South Florida. The graphs show the average values of PM_2.5_ and AOD recorded per month during the period 2012–2016. Note the higher values during summer months.

**Figure 4 jcm-09-02496-f004:**
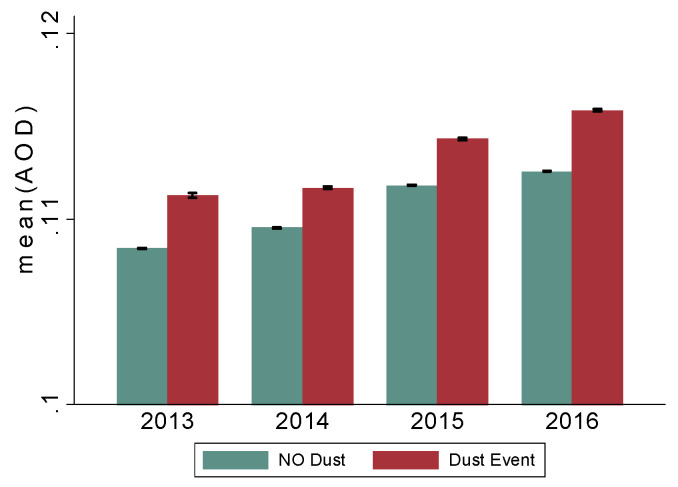
Mean AOD readings in the catchment area 2013–2014 during non-dust and dust events. Note: average AOD readings during dust and non-dust events have been increasing since 2013. AOD: Aerosol Optical Depth (AOD), a proxy of aerosol exposure, was computed at 3 km spatial resolution using satellite-derived MODerate Resolution Imaging Spectroradiometer (MODIS) data.

**Figure 5 jcm-09-02496-f005:**
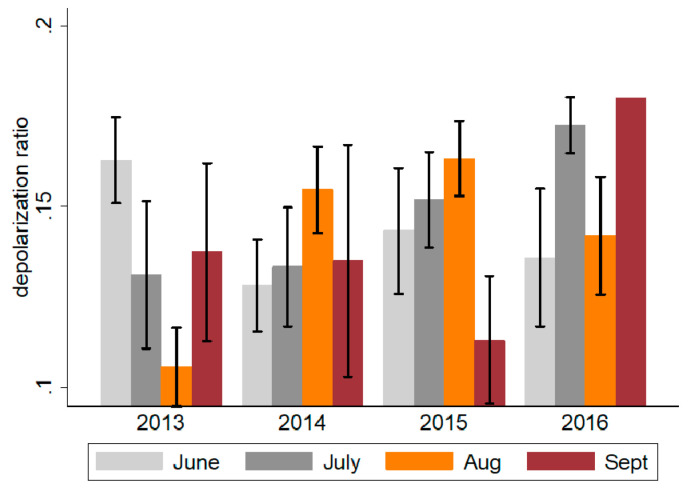
Monthly average depolarization ratio in the Miami catchment area during the summer months, 2013–2016. Obtained using LIDAR, depolarization ratios represent a measure of dust intensity. Values ≤0.1 represent mild dust events, 0.10–0.15 moderate events and ≥0.15 severe events.

**Figure 6 jcm-09-02496-f006:**
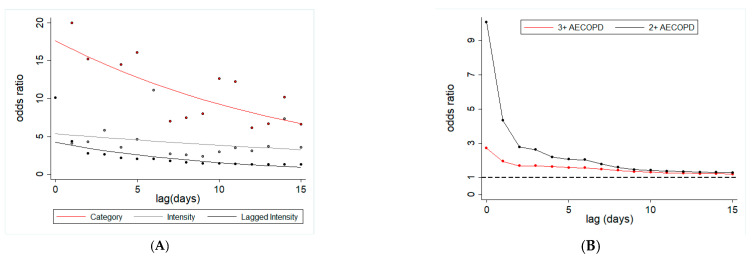
Dust lagged exposure and risk of AECOPD among Miami veterans. (**A**) Odds ratio for two or more AECOPD with respect to three different ways of assessing dust exposure: dust event (yes or no) at a given lag (red line), dust intensity at a given lag (gray line) and cumulative (inverse time weighted) dust event (black line). (**B**) Odds ratio of two or more AECOPD (black line) and three or more AECOPD (red line) with respect to cumulative inverse time weighted dust exposure.

**Table 1 jcm-09-02496-t001:** Empirical association between moderate resolution imaging spectroradiometer (MODIS) AOD (at 3 km spatial resolution) and PM_2.5_ monitored on the ground in South Florida, 2012–2016 (regression coefficient and 95% confidence interval in parenthesis).

Variables	Regression Coefficients (95% Confidence Interval in Parenthesis)
AOD	21.44 ***
	(19.62–23.26)
Season category (0 = October to March, 1 = otherwise)	0.64 ***
	(0.17–1.11)
Factor of Meteorological Conditions (dominant variables atmospheric temperature and dew point)	1.85 ***
	(1.58–2.12)
Factor of Meteorological Conditions (relative humidity)	–0.69 ***
	(−0.89–−0.48)
Constant	7.17 ***
	(6.81–7.52)
Observations	4398
R-squared	0.45

Robust 95% confidence interval in parentheses; *** *p* < 0.01. NOTE: To develop empirical association, AOD data points were collocated with PM_2.5_ monitored on the ground if the time and geographic distance between AOD and PM_2.5_ were ≤0.75 h (or 45 min) and ≤0.025° (~2.5 km), respectively.

**Table 2 jcm-09-02496-t002:** Factor loading of meteorological conditions. Factor 1 and 2 together explained 94% of the total variance.

Variable	Factor 1 *	Factor 2 **	Uniqueness
Wind speed (m/s)	0.00437	−0.03431	0.7825
Ceiling Height (m)	−0.00381	0.02124	0.7487
Visibility (m)	0.00656	−0.03306	0.7531
Dry Bulb Temperature °F	0.60116	−0.62288	0.0286
Dew Point (°F)	0.43451	0.62488	0.0263
Mean sea level pressure (mb)	−0.00650	−0.00624	0.7760
Relative humidity (%)	−0.00762	0.57763	0.0734
% Variance explained	55.46	38.23	

* Factor 1: temperature and dew point high loading along with negative association with atmospheric sea level pressure SLP, ** Factor 2: relative humidity main dominant factor.

**Table 3 jcm-09-02496-t003:** Clinical and demographic characteristics of veterans with COPD studied in Miami between 2013 and 2016.

	ALL	Subjects without AECOPD * (Controls)	Subjects with at Least 1 AECOPD *	*p*-Value
N	296	194	102	
Age (years) ± 95% confidence interval (CI)	70.0 ± 8.7	69.8 ± 8.9	70.3 ± 8.5	0.48
Concomitant asthma (%)	33 (11.1)	13.1	9.4	0.33
Number of exacerbations during 4-year study period ± 95% CI	1.99 ± 3.9	-	2.7 ± 5.0	-
Post bronchodilator lung function values				
FEV1 (L), ± 95% CI	1.6 ± 1.2	1.81 ± 1.7	1.42 ± 0.5	0.015
FEV1 (%), ± 95% CI	46.0 ± 17.0	48.9 ± 18.0	43.2 ±15.5	0.006
FVC (L), ± 95% CI	2.9 ± 0.7	3.1 ± 0.7	2.8 ± 0.7	0.001
FVC (%), ± 95% CI	68.9 ± 16.3	71.3 ± 15.4	66.6 ± 16.8	0.019
FEV1/FVC (%), ± 95% CI	51.4 ± 13.4	52.6 ± 13.6	50.3 ± 13.3	0.15
DLCO (%),± 95% CI	53.1 ± 15.3	53.9 ± 14.7	52.2 ± 15.8	0.37

* AECOPD: acute exacerbation of COPD.

**Table 4 jcm-09-02496-t004:** AECOPD * events recorded among the 296 COPD patients studied in Miami between 2013 and 2016.

GROUP	AECOPD Events	Number of Subjects	Events in Miami Catchment Area		Total
			NO	YES	
Controls	0	194	28	166	194
Cases	1	42	6	36	42
	2	19	4	34	38
	≥3	41	6	227	233
Total		296	44	463	507

* AECOPD: acute exacerbation of COPD.

**Table 5 jcm-09-02496-t005:** AOD, PM_2.5_ and meteorological conditions recorded during dust and non-dust events in the Miami catchment area (2013 to 2016).

	During No Dust Event	Dust Event Intensity *			Total
		≤0.10 (Low)	0.11–0.15 (Medium)	≥0.15 (High)	
Miami Catchment (*n* = 1,322,344)					
Aerosol Optical Depth (AOD)	0.162	0.198	0.246	0.241	0.168
	(0.161–0.162)	(0.196–0.199)	(0.244–0.247)	(0.240–0.242)	(0.168–0.169)
Surface Temperature (°C)	21.0	22.6	22.3	23.1	21.2
	(21.0–21.0)	(22.5–22.6)	(22.3–22.3)	(23.1–23.1)	(21.2–21.2)
Dew Point (°C)	17.6	19.3	18.7	19.9	17.8
	(17.6–17.6)	(19.3–19.4)	(18.7–18.8)	(19.9–19.9)	(17.8–17.8)
Relative Humidity (%)	77.6	78.4	76.9	78.5	77.7
	(77.6–77.6)	(78.3–78.5)	(76.8–76.9)	(78.4–78.5)	(77.6–77.7)
Particulate Matter ≤ 2.5 µm (PM_2.5_) (µg/m^3^)	10.3	12.0	12.9	13.1	10.5
	(10.3–10.3)	(11.9–12.0)	(12.9–13.0)	(13.0–13.1)	(10.5–10.5)
Florida excluding Miami Catchment Area (*n* = 7,648,441)					
Aerosol Optical Depth (AOD)	0.124	0.175	0.201	0.189	0.130
	(0.124–0.124)	(0.175–0.176)	(0.201–0.202)	(0.189–0.189)	(0.130–0.130)
Surface Temperature (°C)	21.0	22.2	21.5	22.4	21.1
	(21.0–21.0)	(22.1–22.2)	(21.5–21.5)	(22.4–22.4)	(21.1–21.1)
Dew Point (°C)	17.6	18.9	18.1	19.2	17.7
	(17.6–17.6)	(18.9–18.9)	(18.1–18.1)	(19.2–19.2)	(17.7–17.7)
Relative Humidity (%)	77.4	78.2	77.7	78.4	77.4
	(77.4–77.4)	(78.1–78.2)	(77.7–77.8)	(78.4–78.5)	(77.4–77.5)
Particulate Matter ≤ 2.5 µm (PM_2.5_) (µg/m^3^)	9.5	11.4	11.7	11.7	9.6
	(9.5–9.5)	(11.4–11.4)	(11.7–11.7)	(11.7–11.7)	(9.6–9.6)
Florida (*n* = 8,970,785)					
Aerosol Optical Depth (AOD)	0.130	0.179	0.209	0.197	0.136
	(0.130–0.130)	(0.178–0.179)	(0.209–0.210)	(0.197–0.197)	(0.135–0.136)
Surface Temperature (°C)	21.0	22.2	21.6	22.5	21.1
	(21.0–21.0)	(22.2–22.2)	(21.6–21.6)	(22.5–22.5)	(21.1–21.1)
Dew Point (°C)	17.6	19.0	18.2	19.3	17.7
	(17.6–17.6)	(19.0–19.0)	(18.2–18.3)	(19.3–19.3)	(17.7–17.7)
Relative Humidity (%)	77.4	78.2	77.6	78.4	77.5
	(77.4–77.4)	(78.2–78.2)	(77.5–77.6)	(78.4–78.5)	(77.5–77.5)
Particulate Matter ≤ 2.5 µm (PM_2.5_) (µg/m^3^)	9.6	11.5	11.9	11.9	9.8
	(9.6–9.6)	(11.4–11.5)	(11.9–11.9)	(11.9–11.9)	(9.8–9.8)

* Estimated with the aid of LIDAR depolarization ratios.

**Table 6 jcm-09-02496-t006:** Average LIDAR * depolarization ratios during dust events and total dust days recorded by month and year in Miami, FL, USA, 2013–2016.

	2013	2014	2015	2016
June	0.16	0.13	0.14	0.14
	(0.15–0.17)	(0.12–0.14)	(0.13–0.16)	(0.12–0.15)
	7 days	16 days	15 days	17 days
July	0.13	0.13	0.15	0.17
	(0.11–0.15)	(0.12–0.15)	(0.14–0.17)	(0.16–0.18)
	9 days	9 days	20 days	23 days
August	0.11	0.15	0.16	0.14
	(0.09–0.12)	(0.14–0.17)	(0.15–0.17)	(0.13–0.16)
	9 days	17 days	9 days	20 days
September	0.14	0.14	0.11	0.18
	(0.11–0.16)	(0.10–0.17)	(0.10–0.13)	(0.18–0.18)
	4 days	6 days	10 days	8 days
Total	0.13	0.14	0.14	0.16
	(0.12–0.14)	(0.13–0.15)	(0.14–0.15)	(0.15–0.16)
	29 days	48 days	54 days	68 days

* LIDAR: Light Detection and Ranging; 95% confidence interval shown in parenthesis. NOTE: Depolarization ratios represent dust intensity: ≤0.10 mild, 0.11–0.15 moderate and ≥0.15 severe dust events.

**Table jcm-09-02496-t007a:** (**a**)

Variables	2+ AECOPD					
	Model 1	Model 2	Model 3	Model 4	Model 5	Model 6
Age (year)	1.002	1.034 ***	1.040 ***	1.001	1.038 ***	1.001
	(0.970–1.035)	(1.009–1.059)	(1.014–1.066)	(0.968–1.035)	(1.012–1.065)	(0.968–1.035)
Asthma status (YES = 1, 0 otherwise)	0.842	1.405	1.6	0.907	1.548	1.123
	(0.264–2.689)	(0.717–2.756)	(0.793–3.224)	(0.278–2.964)	(0.756–3.172)	(0.349–3.611)
Smoking status (1 = Active, 0 otherwise)	2.110 *	1.942 **	1.998 **	2.051	2.007 **	2.112
	(0.891–4.996)	(1.069–3.529)	(1.087–3.671)	(0.849–4.952)	(1.066–3.779)	(0.831–5.367)
Lung function (FEV_1_/FVC)	1.008	1.002	1	1.006	0.998	0.999
	(0.980–1.036)	(0.982–1.021)	(0.981–1.020)	(0.978–1.035)	(0.978–1.018)	(0.972–1.028)
One-day lag PM_2.5_ (µg/m^3^)	1.139 ***			1.124 ***		1.122 ***
	(1.083–1.197)			(1.068–1.184)		(1.061–1.185)
One-day lag ambient temperature (°C)		1.121 ***		1.059	1.103 **	1.062
		(1.033–1.217)		(0.960–1.169)	(1.015–1.198)	(0.961–1.173)
One-day lag dust exposure (LIDAR intensity)			5.777 ***		5.222 ***	4.955 ***
			(2.084–16.010)		(2.077–13.126)	(1.826–13.449)
Constant	0.028 **	0.002 ***	0.022 ***	0.010 ***	0.002 ***	0.011 **
	(0.001–0.525)	(0.000–0.036)	(0.003–0.165)	(0.000–0.327)	(0.000–0.039)	(0.000–0.385)
Observations	169	308	308	169	308	169

Values represent odds ratios and 95% confidence interval is shown in parenthesis; NA = Analysis not applicable because variable(s) was not included in the analysis; *** *p* < 0.01, ** *p* < 0.05, * *p* < 0.1; 95% confidence interval in parenthesis. Since there were gaps in the AOD data, location-specific daily PM_2.5_ measurement was not possible for all days. Thus, the analysis was restricted to observations for all days when PM_2.5_ exposure computation was feasible. Smoking status (active or not) was determined at the time of each independent COPD exacerbation episode by reviewing their medical chart.

**Table jcm-09-02496-t007b:** (**b**)

Variables	≥2 AECOPD					
	Model 1	Model 2	Model 3	Model 4	Model 5	Model 6
Age (year)	0.999	1.034 **	1.038 ***	0.996	1.035 **	0.994
	(0.968–1.032)	(1.005–1.064)	(1.010–1.066)	(0.961–1.032)	(1.006–1.065)	(0.960–1.030)
Asthma status (YES = 1, 0 = otherwise)	1.213	1.931	2.201 **	1.587	2.028 *	1.78
	(0.374–3.937)	(0.869–4.289)	(1.077–4.498)	(0.426–5.910)	(0.903–4.558)	(0.480–6.603)
Smoking status (1 = Active, 0 = otherwise)	1.795	1.882 *	1.784 *	1.749	1.827 *	1.714
	(0.761–4.232)	(0.955–3.709)	(0.953–3.341)	(0.692–4.422)	(0.916–3.644)	(0.678–4.330)
Lung function (FEV_1_/FVC)	1.005	0.998	1.001	1.002	0.997	0.998
	(0.979–1.032)	(0.977–1.020)	(0.981–1.021)	(0.974–1.031)	(0.975–1.020)	(0.970–1.027)
One-day lag PM_2.5_ (µg/m^3^)	1.146 ***			1.111 ***		1.109 ***
	(1.090–1.205)			(1.050–1.175)		(1.045–1.176)
One-day lag ambient temperature (°C)		1.315 ***		1.214 *** (1.096–1.344)	1.301 ***	1.214 ***
		(1.174–1.473)			(1.161–1.457)	(1.096–1.345)
One-day lag dust exposure (LIDAR intensity)			1.741 ***		1.581 **	1.808 **
			(1.176–2.578)		(1.041–2.401)	(1.003–3.261)
Constant	0.025 **	0.000 ***	0.016 ***	0.000 ***	0.000 ***	0.001 ***
	(0.001–0.541)	(0.000–0.001)	(0.002–0.139)	(0.000–0.030)	(0.000–0.001)	(0.000–0.037)
Observations	169	308	308	169	308	169

Values represent odds ratios and 95% confidence interval is shown in parenthesis; NA = Analysis not applicable because variable(s) was not included in the analysis; *** *p* < 0.01, ** *p* < 0.05, * *p* < 0.1; 95% confidence interval in parenthesis. Since there were gaps in the AOD data, location-specific daily PM_2.5_ measurement was not possible for all days. Thus, the analysis was restricted to observation for all days when PM_2.5_ exposure estimates were available. Smoking status (active or not) was determined for the time of each independent COPD exacerbation episode.

**Table 8 jcm-09-02496-t008:** Results of the logistic regression—odds of AECOPD among Miami veterans with COPD with respect to time-lagged dust exposures, 2013–2016.

Lag (day)	Odds Ratio for AECOPD					
	≥2 AECOPD			≥3AECOPD		
	Dust Event (0 = Yes, 1 = No)	Dust Intensity	Distributed Lag	Dust Event (0 = Yes, 1 = No)	Dust Intensity	Distributed Lag
0	NA	10.074 ***	10.074 ***	7.781 ***	2.717 ***	2.717 ***
		(2.241–45.292)	(2.241–45.292)	(2.274–26.621)	(1.460–5.057)	(1.460–5.057)
1	19.930 ***	4.116 ***	4.340 ***	5.859 ***	2.193 ***	1.948 ***
	(2.621–151.548)	(1.700–9.963)	(1.634–11.529)	(1.991–17.242)	(1.373–3.501)	(1.259–3.014)
2	15.162 **	4.269 ***	2.791 **	5.959 ***	2.589 ***	1.684 ***
	(1.895–121.291)	(1.502–12.131)	(1.247–6.247)	(1.675–21.204)	(1.496–4.482)	(1.238–2.291)
3	NA	5.800 ***	2.632 **	21.066 ***	3.603 ***	1.685 ***
		(2.152–15.634)	(1.225–5.654)	(2.850–155.729)	(1.825–7.114)	(1.239–2.292)
4	14.443 ***	3.578 ***	2.201 ***	19.903 ***	3.566 ***	1.639 ***
	(1.921–108.593)	(1.489–8.597)	(1.321–3.667)	(2.649–149.532)	(1.733–7.339)	(1.276–2.106)
5	16.062 ***	4.585 ***	2.065 ***	10.155 ***	3.800 ***	1.586 ***
	(2.154–119.792)	(1.600–13.139)	(1.244–3.430)	(2.259–45.661)	(1.648–8.765)	(1.245–2.021)
6	NA	11.080 ***	2.032 ***	20.757 ***	5.433 ***	1.570 ***
		(2.084–58.915)	(1.247–3.310)	(2.571–167.581)	(1.721–17.153)	(1.229–2.007)
7	7.021 ***	2.714 **	1.781 ***	5.862 ***	2.570 ***	1.490 ***
	(1.611–30.601)	(1.160–6.354)	(1.188–2.671)	(1.661–20.692)	(1.282–5.150)	(1.186–1.873)
8	7.475 ***	2.558 **	1.601 ***	6.226 ***	2.487 ***	1.416 ***
	(1.725–32.395)	(1.218–5.370)	(1.131–2.264)	(1.790–21.654)	(1.316–4.701)	(1.150–1.744)
9	7.956 ***	2.345 ***	1.461 ***	6.646 ***	2.178 ***	1.346 ***
	(1.843–34.346)	(1.272–4.322)	(1.118–1.911)	(1.918–23.037)	(1.250–3.794)	(1.127–1.609)
10	12.621 **	2.936 **	1.416 ***	7.408 **	2.352 **	1.313 ***
	(1.541–103.354)	(1.068–8.070)	(1.124–1.784)	(1.508–36.396)	(1.080–5.119)	(1.118–1.543)
11	12.186 **	3.477 **	1.376 ***	4.442 **	2.249 **	1.277 ***
	(1.558–95.288)	(1.058–11.423)	(1.117–1.695)	(1.160–17.019)	(1.100–4.597)	(1.105–1.476)
12	6.153 **	3.127 **	1.343 ***	4.986 **	2.502 **	1.256 ***
	(1.393–27.186)	(1.284–7.614)	(1.112–1.623)	(1.389–17.897)	(1.184–5.288)	(1.096–1.440)
13	6.635 **	3.660 **	1.315 ***	5.409 ***	2.885 **	1.237 ***
	(1.507–29.212)	(1.357–9.868)	(1.105–1.566)	(1.515–19.309)	(1.269–6.560)	(1.086–1.409)
14	10.165 **	7.322 **	1.300 ***	6.164 **	3.366 **	1.222 ***
	(1.317–78.442)	(1.171–45.791)	(1.103–1.533)	(1.314–28.903)	(1.194–9.487)	(1.079–1.384)
15	NA	3.553 **	1.282 ***	5.495 ***	2.823 ***	1.208 ***
		(1.343–9.400)	(1.090–1.507)	(1.566–19.280)	(1.291–6.173)	(1.069–1.364)

AECOPD: Acute exacerbation of COPD; NA = could not be estimated; 95% confidence interval in parenthesis; *** *p* < 0.01, ** *p* < 0.05.

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
