# Peer review of "Association between African Dust Transport and Acute Exacerbations of COPD in Miami"

_jcm, 2020, doi:10.3390/jcm9082496_

Round 1
Reviewer 1 Report
The authors provide an interesting analysis looking at COPD exacerbations in a panel of COPD patients followed at the Miami VA hospital in relation to air pollution episodes from dust clouds. They find a relation between 2+ AECOPD and this dust related air pollution that lasted up to 15 days.
Given the recent reports on Saharan dusts storms- this is timely .
Comments:
The air pollution piece of the paper is very elegantly presented- but a bit beyond my main area of expertise.
The AECOPD piece was a bit confusing. For example- they list patients as "Cases" and "Controls" , with cases having 2+ events, yet in Table 4 even the " controls" with 0 events had 166 AECOPD? This makes no sense to me and needs to be clarified.
Typically, in these type of panel studies events are tracked and reported. Thus, the figure 2 graphic would have an additional row for AECOPD ( with the number of events noted on the y axis). My interpretation of the results is that the outcome evaluated was the number of days where there was at least 2 AECOPD ( regardless of whether people came from the case or control group). The data in Table 4, suggesting that even controls have AECOPD, makes me think this is the case- but some clarification is needed.
Author Response
Reviewer 1
The authors provide an interesting analysis looking at COPD exacerbations in a panel of COPD patients followed at the Miami VA hospital in relation to air pollution episodes from dust clouds. They find a relation between 2+ AECOPD and this dust related air pollution that lasted up to 15 days.
Given the recent reports on Saharan dusts storms- this is timely.
AR: Thank you for recognizing the timeliness of the paper.
Comments:
The air pollution piece of the paper is very elegantly presented- but a bit beyond my main area of expertise.
The AECOPD piece was a bit confusing. For example- they list patients as "Cases" and "Controls" , with cases having 2+ events, yet in Table 4 even the " controls" with 0 events had 166 AECOPD? This makes no sense to me and needs to be clarified.
AR: We apologize for the confusion caused by formatting issue that the line was shifted to show 1 event under control. All subject who did not have any AECOPD were control and the rest were events. So, the events modelled with respect to controls. Now, the corrected table shows controls and number of events with the unique number of patients under each event.
Typically, in these type of panel studies events are tracked and reported. Thus, the figure 2 graphic would have an additional row for AECOPD ( with the number of events noted on the y axis). My interpretation of the results is that the outcome evaluated was the number of days where there was at least 2 AECOPD ( regardless of whether people came from the case or control group). The data in Table 4, suggesting that even controls have AECOPD, makes me think this is the case- but some clarification is needed.
AR: You are correct, issue was caused due to wrong formatting of the table 4. We do report number of events and number of unique patients under the frequency of events.
Reviewer 2 Report
In this study, authors presented link between African dust transport and exacerbations of chronic obstructive pulmonary disease. Authors adopted light detection and ranging and satellite-based aerosol optical depth methods in order to identify the dust event and consequence particulate matter exposure. Authors conducted detailed analysis of extracted data and concluded that Saharan dust outbreaks elevates PM concentration and risk of acute exacerbation of COPD. Authors provided in-depth discussion regarding the topic.
Minor edit required:
- Line 32: What is the relationship between Saharan dust outbreaks and ambient temperature exposure? In result and discussion authors also discuss relationship between AECOPD and ambient temperature.
- Figure 3. Label on Y-axis missing or not clear
- Figure 4 and 5. Label on Y-axis missing or not clear
- Figure 6B. Label on Y-axis missing
Author Response
Thank you for the opportunity to revise our manuscript. We have carefully read reviewers’ comments and addressed them in the revision. Below we provide point by point response their comments as well. Each response is blue color and begin with AR:
Reviewer 2
In this study, authors presented link between African dust transport and exacerbations of chronic obstructive pulmonary disease. Authors adopted light detection and ranging and satellite-based aerosol optical depth methods in order to identify the dust event and consequence particulate matter exposure. Authors conducted detailed analysis of extracted data and concluded that Saharan dust outbreaks elevates PM concentration and risk of acute exacerbation of COPD. Authors provided in-depth discussion regarding the topic.
AR: Thank you
Minor edit required:
Line 32: What is the relationship between Saharan dust outbreaks and ambient temperature exposure? In result and discussion authors also discuss relationship between AECOPD and ambient temperature.
AR: Yes, the ambient temperature was slightly higher during dust events (see lines 176-178). It is not because of dust events per se, but because most dust events are in summer and early fall. But when we restrict the analysis during summer and fall, the ambient temperature did not vary significantly between dust and non-dust event days.
Figure 3. Label on Y-axis missing or not clear
AR: We have regenerated figure. BTW it represents PM2.5 concentration (ug/m3)
Round 2
Reviewer 1 Report
My prior concerns have been addressed